# From Bench to Bioactivity: An Integrated Medicinal Development Based on Kinetic and Simulation Assessment of Pyrazolone-Oxadiazole Coupled Benzamide as Promising Inhibitors of Diabetes Mellitus

**DOI:** 10.3390/ph18111595

**Published:** 2025-10-22

**Authors:** Manal M. Khowdiary, Shifa Felemban

**Affiliations:** Department of Chemistry, Faculty of Applied Science, University College-Al Leith, University of Umm Al-Qura, Makkah 21955, Saudi Arabia

**Keywords:** oxadiazole, diabetes mellitus, simulation, enzyme kinetics, pharmacophore

## Abstract

**Background**: In this research work, novel pyrazolone-derived oxadiazole-based benzamide derivatives **(1–10)** were synthesized through unique and facile synthetic routes. Introduction: These scaffolds were designed to be therapeutically more effective and have fewer side effects. **Methods**: To confirm the structure of analogs in detail, we employed ^1^HNMR, ^13^CNMR, and HREI-MS spectroscopy. The potential of all derivatives was tested by screening them against alpha-amylase and alpha-glucosidase in comparison with reference anti-diabetic drug acarbose (4.50 ± 0.20 µM and 4.90 ± 0.30 µM). **Results & Discussion**: Among all tested analogs and standard drugs, derivative 3 proved to be the most promising candidate. It exhibited the most powerful inhibitory effect (IC_50_ = 3.20 ± 0.20 µM and 3.60 ± 0.10 µM). To further investigate its activity, the experimental results were supported by in silico investigations. Molecular docking demonstrated strong and viable interactions between enzymes and the most potent compound. DFT calculations validated the electronic configuration, stability, and reactivity of lead molecules. Furthermore, the ADMET profile predicted the favorable drug likeness properties and low toxicity. The results of docking were further confirmed via molecular dynamics analysis, whereas the pharmacophore model of analog **3** supports the formation of a stable hydrogen bond network of derivatives with the receptor site of the enzyme. **Conclusions**: Collectively in silico and in vitro results underscore the therapeutic potential of these derivatives for the effective treatment of diabetes in the future.

## 1. Introduction

Diabetes mellitus (DM) is a chronic metabolic disorder characterized by persistent hyperglycemia resulting from impaired insulin secretion or action. It ranks among the top global health challenges and contributes to severe complications such as neuropathy, nephropathy, retinopathy, and cardiovascular diseases [1,2,3]. Type 2 DM (T2DM) constitutes about 90% of all cases, and its prevalence continues to rise worldwide, posing a major threat to public health and socioeconomic systems [4].

Although several synthetic oral agents, such as sulfonylureas, biguanides, and α-glucosidase inhibitors, are available for the management of diabetes, their long-term administration is often associated with adverse effects including hypoglycemia, gastrointestinal irritation, dizziness, and unwanted weight gain [5,6]. These safety concerns necessitate the search for novel, potent, and well-tolerated antidiabetic agents. Natural bioactive compounds, such as flavonoids, polyphenols, and alkaloids, have shown promising potential as safer enzyme inhibitors; however, challenges remain in their stability, selectivity, and pharmacokinetics [7].

Postprandial hyperglycemia, a key hallmark of T2DM, is primarily regulated by carbohydrate-hydrolyzing enzymes such as α-amylase and α-glucosidase [8]. α-Amylase, produced by the pancreas and salivary glands, catalyzes the breakdown of starch into smaller sugars like maltose, while α-glucosidase in the intestinal lining converts disaccharides into absorbable monosaccharides such as glucose [9,10]. Excessive activity of these enzymes accelerates glucose absorption, leading to elevated blood sugar levels. Hence, dual inhibition of α-amylase and α-glucosidase represents a well-validated therapeutic strategy for maintaining glucose homeostasis [11].

Heterocyclic compounds play a crucial role in modern drug design due to their structural diversity and biological versatility [12]. Among them, 1,3,4-oxadiazole scaffolds have gained attention for exhibiting a wide spectrum of pharmacological properties, including antidiabetic, antibacterial, anti-inflammatory, and carbonic anhydrase inhibitory effects [13,14,15,16]. However, compared to extensively studied 1,3,4-thiadiazole derivatives, oxadiazole–sulfonamide hybrids remain underexplored for enzyme-targeted diabetes therapy [17]. Incorporation of oxadiazole and sulfonamide moieties within a single molecular framework may enhance target selectivity, hydrogen-bonding capacity, and metabolic stability, thereby improving biological activity [18,19].

Based on these considerations, the present study was designed to synthesize and evaluate novel thiadiazole-bearing sulfonamide derivatives as potential dual inhibitors of α-amylase and α-glucosidase. Computational techniques including molecular docking and ADMET analysis were applied to elucidate the binding interactions, predict pharmacokinetic behavior, and establish structure–activity relationships (SARs). The overarching hypothesis proposes that hybridizing thiadiazole and sulfonamide pharmacophores can yield compounds with enhanced inhibitory potency, drug-likeness, and safety for effective diabetes management [18].

The rationale of the present study (Figure 1) involves a comparative analysis between the newly synthesized analogs (1–10) and previously reported oxadiazole-based derivatives [20,21]. This comparison was undertaken to elucidate the structure–activity relationship (SAR) and to understand how specific structural modifications influence the biological potency and overall efficacy of the designed compounds.

## 2. Results and Discussion

### 2.1. Chemistry

The first step consists of mixing small amount of carbon disulfide with potassium hydroxide in ethanol. This reaction mixture was combined with amino substituted pyrazolone compound (I), and the mixture was refluxed for about 3 h, with constant stirring. This produced the first intermediate, pyrazolone, with hydrazine carboxamide group (II). In the secondstep, intermediate (II) was cyclized by reacting it with 1,4-dioxane (cyclizing agent) and hydrazine carboxamide, using triethylamine as a base. Following a reflux period of 14 h under constant magnetic stirring, it yielded a second intermediate, an amino-based oxadiazole with pyrazolone group (III). By using ethanoic acid as a catalyst and ethanol as a reaction medium, intermediate (III) was reacted with substituted benzoyl chloride; after refluxing for 4 h, the desired pyrazolone-derived oxadiazole-based benzamide derivates **(1–10)** were obtained with good yields (Figure 1). The desired products were washed by using n-hexane to remove the impurities left behind, and the final product was dried to obtain it in pure form. The reaction progress and intermediates were checked regularly using TLC.

### 2.2. Biological Activities (SAR)

After successfully synthesizing the pyrazolone-derived oxadiazole-based benzamide compounds **(1–10)**, their biological activities were evaluated to determine how well they inhibited the target enzymes. Various inhibitory potency levels were displayed by all ten analogs, ranging from 5.30 ± 0.10 µM to 3.90 ± 0.20 µM (alpha-amylase) and 5.90 ± 0.20 µM to 4.10 ± 0.10 µM (alpha-glucosidase), as shown in Table 1. The majority of the drugs showed excellent inhibition compared to the commercially available drug acarbose with an inhibition potential from 4.50 ± 0.20 µM to 4.90 ± 0.30 µM for both enzymes. The inhibition power of derivatives depends upon various factors like size, position on varied substituted phenyl ring, nature, and capability to create polarity on ring. The basic structure of the newly synthesized compound is shown in Figure 2**.**

We have successfully prepared ten anti-diabetic analogs in good amounts. These compounds were tested for their ability to block specific enzymes, and most of them showed moderate to strong inhibition, significantly disrupting the enzyme activity.

Among the hydroxyl group-substituted analogs, analog **3** had a potency of 3.20 ± 0.20 µM (α-amylase) and 3.60 ± 0.10 µM (α-glucosidase). It exhibited the highest anti-diabetic activity and became the lead compound of the series. It might be due to the occupation of the most attractive and electron-releasing (-OH) *ortho* and *para* carbons of the benzene ring. These substituents have an electron-releasing nature due to a lone pair of electrons and polarity on the ring. This creation of polarity helps analog **3** bind tightly to the enzymes, slowing down the sugar breakdown. The second most potent was analog **10**, with a potency of 3.90 ± 0.20 µM and 4.10 ± 0.10 µM, showing excellent inhibition. It might be due to the residence of the hydroxyl group at *ortho* and *meta* carbons, as presented in Figure 3. This indicates that modifying the number and position of groups on a substituted benzene ring alters the binding strength of the analogs. In simple terms, the activity of a derivative depends on where and how functional groups are attached, making some analogs more effective than others in the series.

By studying the activities of analogs **5** and **7,** analog **5** with inhibition power (5.40 ± 0.10 µM and 6.10 ± 0.20 µM) and analog **7** with potency (4.10 ± 0.30 µM and 4.60 ± 0.20 µM) showed very good inhibitory power by engaging the enzyme with itself. This might be due to the occupation of the hydroxyl group at the *meta* carbon in molecule **5** and substitution of the active fluorine atom at the *para* and hydroxyl group at the *meta* carbon in derivative **7,** as given in Figure 4. Molecule **7** is more potent than molecule **5** because it contains a highly electronegative fluorine (-F) atom at the *para* carbon of the ring. This fluorine atom withdraws pi electrons from the aromatic ring, creating a partial positive charge. As a result of this, the molecule forms stronger hydrogen bonds with the enzyme active sites, enhancing its inhibitory effect.

Analog **4** has a fluorine atom at the *para*-position and methyl group at the *meta*-position (5.10 ± 0.20 µM and 5.90 ± 0.30 µM), and analog **1** with a *para*-fluoro substitution has a biological potential of 5.30 ± 0.10 µM and 5.90 ± 0.20 µM. Analog **4** is more potent than analog 1, and one reason for this is that analog **4** has most attractive and electron withdrawing fluorine atom, which due to its small size and high electronegativity, attracts the electrons from the ring and enables the derivative to engage with the enzymes, as shown in Figure 5. In analog **1** with the fluorine atom, we can see the attachment of a (-CH_3_) group, which did not engage analog **1** to make strong attractions with the respective enzyme due to a large-sized methyl group. This shows us that the size of attached substituents also play a major role for making a derivative potent inhibitor. Hence, there is a slight decrease in the binding potency of derivative **1**.

In the case of halogen-substituted analog **8** (11.70 ± 0.20 µM and 12.10 ± 0.10 µM) and analog **9** (15.30± 0.10 µM and 16.10 ± 0.20 µM), the activity of molecule **8** is better than molecule **9** for both enzymes. Analog **8** has a well-defined electronegative chlorine atom, which increases its binding capabilities. Moreover, the methyl group at the *meta*-position on the same analog also enhanced its activity possibly due to optimal positioning. Analog **9** displayed weak inhibitions towards targeted enzymes. It might be because of the large size bromine atom, which is less electronegative than the chlorine atom and hinders strong attractions with the given enzymes (Figure 6).

The biological potential of analog **6** is 14.60 ± 0.20 µM and 15.10 ± 0.50 µM with the *para*-substituted cyanide group (-CN), and that of derivative **2** is 10.50 ± 0.20 µM and 11.20 ± 0.10 µM with *meta*-substituted (-CH_3_) groups. This difference in potency likely occurs because the (CH_3_) at the *meta*-position creates more favorable binding forces compared to the *para*-positioned -CN group in analog **6**, as presented in Figure 7. The -CN group electronic effects and steric positioning at the *para* carbon may be less optimal for strong binding, resulting in reduced inhibitory efficiency.

### 2.3. Kinetics Study

Enzyme kinetics studies provide the most vital information for how drugs work against the disease, such as binding of the inhibitor with its target, and the concentration required to achieve the desired effects. This study explains that the amount of the drug will be affected, slowing down the biological process. The different patterns for slowing down a process are obtained in the form of curves or slopes. This also provides how various forms of inhibitors or derivatives cause the reaction kinetics to slow in unique ways by adding more inhibitors. In this study, analog **3** displayed outstanding inhibition potency in the form of competitive inhibitors. The substitution of unique functional groups in various sites of the phenyl ring might be responsible for its remarkable efficiency. The graphical curves in Figure 8 show the direct competition of the analog against the substrate for the same spot on the enzyme. The decrease in both attraction for its natural substrate (Km) and efficiency of the enzyme (Vmax) was observed. The graph also shows a bunch of lines that point downward, but their point of origin on the y-axis is similar, which is transferred from a higher to lower value. It means that the drug binds itself with the active site of the enzyme, and this attachment makes the enzyme less effective and less likely to bind with its natural substrate. Figure 9 and Figure 10 show us that by adding the inhibitor (drug), the extent of inhibition in the form of % inhibition along the y-axis increases, and it became its peak value of 50% inhibition. After that, by adding more inhibitor, there is no noticeable change in the inhibition percentage, showing the maximum inhibition against the specific disease-causing enzyme.

### 2.4. Docking Analysis (In Silico Approach)

A molecular docking study of the most potent molecules, **3** and **10**, was carried out to investigate the attractions of enzymes (proteins) with ligands (derivatives). This computational study helps to predict the best fit or most stable orientation of a small molecule within the binding site of large molecules. It also helps to identify the potential drug candidate by screening a large library of molecules and predicting their binding with the target protein. The rapid screening of many compounds significantly saves time and a massive amount of money, which is associated with normal approaches of drug discovery. By understanding these interactions, researchers can modify its structure to improve its binding affinity and drug efficacy. Moreover, it explores different possible positions or poses of the compound with the receptor site of the enzyme. Docking software PyMOL 3.1.6.1, AutoDock 4.2.6, and DSV 2021 (v21.1.0.20298) were used [22,23,24]. Docking analysis of top most compounds 3 and 10 revealed that while binding with the enzyme, it does not causes any abnormal conformational changes in the enzyme substrate complex, displays a stable and effective binding attraction, and causes an outstanding inhibition against both mentioned enzymes. Moreover, it also confirms the existence of very strong intermolecular forces, which these analogs developed. It may be due to the presence of hydrogen bond-making groups *(-OH)* located at different positions on the aromatic ring in both molecules. Effective *2D* and *3D* interactions of the enzyme with proteins of lead compound **3** are GLU-1-199, ASP-232, and SER-A-200, which are conventional hydrogen bonds and interactions exhibited by oxadiazole rings TYR-A-334 and PHE-A-330 Pi-Pi, as displayed in Figure 11 and Figure 12 and analog **10** presented in Appendix A.

### 2.5. Molecular Dynamics Simulation (MD)

For deep analysis of how the enzyme and derivative interact with each other, a molecular dynamic simulation study was conducted. This computer-based approach tracks the movement and behavior of atoms and molecules over time. As discussed earlier, molecular docking helps identify the binding interactions between our derivative and the enzyme’s receptor site. Meanwhile, molecular dynamics (MD) simulations track how this complex moves and interacts over time. Through this study, we observed the structural changes induced by the ligand when binding to the enzyme’s receptor site. Additionally, these computer simulations helped confirm the stability of these interactions, giving us deeper insight into the drug’s potential behavior. To assess the stability of the protein–ligand complex, we tracked its movements over a 100-nanosecond (ns) simulation period. Employing Root Mean Square Deviation (RMSD) analysis, we measured structural fluctuations and confirmed the overall stability of the complex throughout the simulation analysis. However, MD study of lead analog **3** proved that no major or unstable structural changes caused by the analog with its respective enzymes and interactions was stable enough, which might be due to the substitution of hydroxyl groups at different sites (*ortho* and *para*) on the aromatic ring. Figure 13 represents the MD simulation of our lead compound **3** of the series.

The root mean square fluctuation (RMSF) plot (Figure 14) illustrates the flexibility of individual amino acid residues within the protein–ligand complex throughout the molecular dynamics (MD) simulation. RMSF values, expressed in Ångströms (Å), indicate the average positional deviation of each residue from its mean position during the trajectory. Lower RMSF values represent more rigid and stable regions, typically corresponding to the protein’s core or secondary structural elements (α-helices and β-sheets), whereas higher RMSF values reflect more flexible regions, such as surface loops or terminal residues.

In the present study, most residues exhibited low RMSF values (<2 Å), indicating the overall structural stability of the protein upon ligand binding. Only minor fluctuations were observed at loop regions, suggesting that the ligand interaction does not induce significant conformational instability. The consistent low fluctuations in active-site residues further confirm the stable binding of the ligand within the catalytic pocket during the simulation period. Figure 14 represents the RMSF of our lead compound **3** of the series.

The hydrogen bond occupancy graph (Figure 15) represents the percentage of time-specific hydrogen bonds maintained during the molecular dynamics (MD) simulation of the protein–ligand complex. Each bar on the x-axis corresponds to a particular residue in the protein that participated in hydrogen bonding with the ligand, while the y-axis shows the occupancy value, typically ranging from 0.00 to 0.07 (or 0–7%) in this figure. A higher bar indicates a hydrogen bond that persisted for a greater proportion of the simulation time, reflecting a more stable and consistent interaction.

From the graph, it is evident that residues at the beginning of the sequence (left side, e.g., residue indices 1–1300) exhibit a higher hydrogen bond occupancy, suggesting they play a key role in stabilizing the ligand within the binding pocket. In contrast, residues towards the later region (beyond residue 3000) show a much lower occupancy, implying that hydrogen bonding interactions in those regions are either transient or absent.

Overall, this analysis demonstrates that the ligand forms stable and sustained hydrogen bond interactions with specific active site residues throughout the simulation period, supporting its strong binding affinity and stability within the protein’s active site.

### 2.6. Pharmacophore Modelling

The 3D shape of a molecule is important for interacting with its target-like enzyme. This shape consists of some key chemical features including hydrogen bond donors (HBD) and acceptors, hydrophobic regions, aromatic rings, and charged groups. To visualize these critical binding interactions and deep interpretations of the hydrogen bond network inside the complex, we created a pharmacophore model of our most potent inhibitor from the series. This model clearly displays all of these essential binding features and their 3D arrangement in space. One of the main applications of this tool is screening chemical libraries to identify potential drug candidates. It also helps medicinal chemists to understand structure–activity relationships (SARs) and how a derivative structure affects its biological activity. The study also reveals the strength and nature of binding interactions between the molecule and its targeted enzyme. Compounds that form extensive hydrogen-bonding networks generally demonstrate stronger binding affinity to the given enzyme, marking them as promising lead candidates for further development of drugs. Pharmacophore modelling of our top most analog **3**, as presented in Figure 16 and Figure 17, and analog **10** in Appendix A shows that the (-OH) group located at *ortho* and *para* carbons of varied substituted aromatic rings gives it a more stable and stronger hydrogen bond ability compared to the other members of the same series. This means that this molecule can bind more effectively and efficiently with its biological target and can act as a promising drug candidate for diabetes.

### 2.7. MESP

The molecular electrostatic potential is a useful tool for understanding the electronic structure and chemical properties (reactivity) of molecules. It helps identify which parts of a drug are likely to attract electrophiles (electron-seeking groups) or nucleophiles (electron-donating groups). The low electronic region inside a molecule called σ-holes and π-holes are responsible for the formation of non-covalent bond interactions when the positive part of one molecule is attracted by the negative part of another molecule [25]. It also helps in designing new drugs by showing where a drug molecule is likely to interact with its specific target, like a protein based on its electrostatic potential. Moreover, it also visualizes the distribution of the charge in a molecule and its effects on other molecule, which could lead to the development of more potent medicines [26,27,28]. These charge differences help to explain how non-covalent bonds form and stabilize this interaction. By mapping the molecule’s charge distribution, MESP shows how the drug might interact with its target enzyme. Areas with high electron density (shown in red on MESP maps) are more likely to undergo electrophilic attacks, while electron-deficient regions (blue areas) attract nucleophiles. This makes MESP valuable for understanding and improving drug design.

### 2.8. FMO Approach

A molecule’s reactivity is determined by its electronic environment and polarity, which influence how it interacts with other substances. The HOMO and LUMO orbitals are key to understanding how a molecule behaves. The HOMO is the highest and stable energy level where the electronic density is present, while the LUMO is the lowest or unstable empty level that is unoccupied by electrons [29,30]. The difference in energy between these two levels is called the HOMO-LUMO gap and affects how stable or reactive the molecule is. A smaller gap shows that the molecule is more reactive as electrons can move more easily between orbitals, whereas a larger gap indicates greater stability [31,32]. The energy difference between two orbitals in our most potent compound **3** is 0.3339 eV, which represents its extra stability and potential binding capabilities with the targeted enzyme. The color bar on the left side of Figure 18 in the form of different colors actually represent the presence or absence of electrons, with the bright red color showing the strong nucleophilic sites and the bright blue color showing the electron deficient sites, such as electrophilic sites. Understanding these properties helps predict how a molecule will behave in chemical reactions, particularly in drug–target interactions. In order to understand the electronic parameters of our top most inhibitor **3**, the series of pyrazolone-derived oxadiazole-based benzamide derivatives are displayed in Figure 18 and analog **10** in Appendix A.

### 2.9. ADMET Investigation

Researchers employ computational tools (in silico ADMET) to forecast the pharmacokinetics parameters of drugs during the early stages of development. The expansion of safe and effective drugs is a complex process that requires a deep understanding of how a drug interacts with the human body. The process of making new drugs involves many steps leading from its initial phase of development to preclinical testing and extensive clinical traits. ADMET properties are essential for assessing the pharmacokinetics and pharmacodynamics of a drug candidate. These properties consist of absorption, a process in which the drug enters the bloodstream, a critical factor that directly affects its bioavailability. Once the drug enters into the bloodstream, its distribution explains how it travels throughout the body. Understanding its distribution is vital because it effects its concentration at the target site and potential interactions with other cells and tissues. The metabolism process helps the body chemically alter the drug and facilitate its elimination. Through excretion, the body eliminates the drug, and toxicity assessment is a critical aspect that involves the potential toxic effects of the drug in living organisms. These parameters play a crucial role in determining the overall quality and safety of a drug, whether the drug is safe to use or not. In this research study, ADMET properties of analogs **3** and **10** were thoroughly examined by using important criteria. The analysis included skin permeability (log K_p_), “Lipinski’s rule” to assess drug-likeness, and PAINS filters to rule out problematic compounds. Additionally, Muegge’s criteria were applied for lead optimization, while lead-likeness was evaluated to check their suitability for further development as a drug. Other key factors like Egan’s bioavailability predictions, Ghose, Brenk, and Veber’s drug-like property rules, and synthetic feasibility (how easily the compounds can be produced) were also considered. These comprehensive tests help explain that our top most analogs, **3** and **10**, do not act against any of the rules mentioned above, and they can act as a potential and therapeutically safe anti-diabetic agent in the future. Figure 19 represents the safety level and characteristic properties of compound 3, and Appendix A shows analog **10**. Moreover, the web diagram in Figure 20 and Appendix A further enhances the ADMET properties like the molecular weight, hydrogen bond donor, and acceptor site of top ranked compounds. A comprehensive analysis of drug likeness properties is presented in Appendix A.

## 3. Experimental

### 3.1. General Information

After the successful synthesis of our target product, we determined their molecular masses and formulas by using a high resolution (HR-MS) mass spectrometer. Besides finding their exact masses it also explored their accurate pattern of fragmentation. All chemicals and solvents were taken from authentic suppliers (Sigma-Aldrich, Merck, Germany) and were highly pure to keep reactions running smoothly. We checked the reaction progress and formation of the intermediate in each step in the form of a clear spot by using TLC with silica gel plates on aluminum backing. The TLC plates were examined under U.V. light under a wavelength of 254 nm to 365 nm to visualize the compounds. We also measured the melting and boiling points precisely with a Büchi M-360 device to confirm that the obtained molecule was free from impurities and met stability criteria. Structural confirmation, number and nature of peak like singlet, doublet, and triplet were determined using spectroscopic tools like HNMR, and spectra were acquired at 600 MHz. To understand the carbon environment or number of carbon atoms in compounds and their connectivity, we used ^13^CNMR and collected data at 150 MHz. Chemical shifts (difference in resonance frequency of a proton and that of standard) δ are listed in parts per million (ppm) compared to the solvent (medium) peak, and coupling constants *(J)*, which indicates the strength of the proton interactions, are given in hertz *(Hz)*. For these tests, DMSO-***d_6_*** was employed as both the solvent (medium) and reference standard.

### 3.2. Synthetic Methodology

In the first step, 2 mL of carbon di-sulfide was mixed with alcoholic potassium hydroxide solution (10 mL). This reaction mixture was combined with 2 mmol of amino substituted pyrazolone compound (I), and the mixture was refluxed for about 3 h with constant stirring. This produced the first intermediate pyrazolone with hydrazine carboxamide group (II).

In the secondstep, intermediate (II) was cyclized by reacting with 10 mL of 1,4-dioxane (cyclizing agent) and hydrazine carboxamide, using triethylamine as a base (2–3 drops). Following reflux for 14 h, it produced the second intermediate, an amino-based oxadiazole with a pyrazolone group (III).

By using ethanoic acid as a catalyst and ethanol as a reaction medium (10 mL), intermediate (III) was reacted with different substituted derivatives (2 mmol each) after refluxing for 4 h; the desired pyrazolone-derived oxadiazole-based benzamide derivates **(1–10)** were obtained in good yields (Figure 1). The final product was washed with *n*-hexane to remove the impurities left behind, and dried the final product to obtain it in pure crystalline form. The reaction progress and intermediates were checked regularly using TLC.

### 3.3. Protocols

Assay protocols including docking, DFT, alpha-amylase, alpha-glucosidase, MD simulations, enzyme kinetics study, and spectral analysis of synthesized molecules are provided separately in the Appendix A.

## 4. Conclusions

This research study focused on the synthesis of pyrazolone-derived oxadiazole-based benzamide derivatives **(1–10)**, and to confirm their structure in detail, we employed ^1^HNMR, ^13^CNMR, and HREI-MS spectroscopy. After structural confirmation, the compounds were tested for anti-diabetic activity, with most showing moderate to strong inhibition. Among them, molecules **3, 7**, and **10** exhibited significantly higher activity than the reference drug acarbose. This might be due to the occurrence of electron-releasing hydroxide (-OH) groups and halogen-like (-F) atom at varied substituted phenyl rings. To figure out how these compounds bind to the target protein, we used molecular docking simulations. The results showed strong stable interactions, explaining their high binding affinity. Further analysis using Density Functional Theory (DFT) helped explain their strong inhibitory effects and their electronic structures. MD simulations analysis is used to investigate the dynamic movement and stability of the enzyme–substrate complex. Moreover, the pharmacophore model was generated to study and confirm the wide and stable hydrogen bond network of lead compound **3**. An ADMET study was also implemented on the most potent analog to explain the safety level and pharmacokinetic properties of the drug, which will allow us to use these active molecules as a strong inhibitor of diabetes in the future.

## Data Availability

The raw data supporting the conclusions of this article will be made available by the authors on request.

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
