# Peer review of "From Bench to Bioactivity: An Integrated Medicinal Development Based on Kinetic and Simulation Assessment of Pyrazolone-Oxadiazole Coupled Benzamide as Promising Inhibitors of Diabetes Mellitus"

_pharmaceuticals, 2025, doi:10.3390/ph18111595_

Round 1
Reviewer 1 Report
Comments and Suggestions for Authors
The manuscript entitled "From Bench to Bioactivity: An Integrated Medicinal Development Based on Kinetic and Simulation Assessment of Pyrazolone-Oxadiazole Coupled Benzamide as Promising Inhibitors of Diabetes Mellitus" requires substantial improvements to the text (clarity, flow, and organisation) and to the experimental methods (purification/ purity criteria, complete spectral datasets, and consistency of the Supplementary Information). Therefore, I recommend consideration for publication in "Pharmaceuticals" with the following major revisions:
- In Section S1. Spectral analysis, (Supplementary information) please add the physicochemical properties for each synthesized compound (physical state : powder or oil, colour, and melting point). Also, after each compound's name, include its compound number as used in the manuscript.
- In Section S1. Spectral analysis (Supplementary Information), please include also the spectral data and the physicochemical properties for the three intermediates used to synthesize the final compounds (Intermediates I, II, and III).
- The 1H and 13C NMR spectra for Intermediates I-III and Compounds 4, 5, 6, 7, 8, and 10 are missing. Please include them in the Supplementary Information.
- Several figures display inconsistent sizing. Please standardise figure dimensions across the manuscript to ensure a uniform presentation.
- The Introduction would benefit from a focused rewrite. Please streamline to 4-5 concise paragraphs with a coherent progression (clinical need ‚ therapeutic limitations‚ enzyme-based rationale‚ scaffold rationale‚ study aims). Clearly articulate the research gap and novelty versus prior oxadiazole reports, and strengthen the chemical rationale (structure-property/activity arguments, with targeted citations). Conclude with a succinct objective and hypothesis that previews the assays and SAR to be tested.
- The statement in section 3. Experimental ("washed with straight-chain hydrocarbon hexane…"), does not sufficiently describe how the products were purified or how purity was established. Kindly detail the purification workflow for each compound (e.g., silica gel flash chromatography with gradient n-hexane/EtOAc or recrystallization from EtOH, etc.)
Author Response
- In Section S1. Spectral analysis, (Supplementary information) please add the physicochemical properties for each synthesized compound (physical state : powder or oil, colour, and melting point). Also, after each compound's name, include its compound number as used in the manuscript.
Reply: Physiochemical properties of all the synthesized compounds including physical state : powder or oil, color, and melting point were added in supplementary information file. Moreover, compound number was also added as per kind suggestion.
- In Section S1. Spectral analysis (Supplementary Information), please include also the spectral data and the physicochemical properties for the three intermediates used to synthesize the final compounds (Intermediates I, II, and III).
Reply: spectral data and the physicochemical properties for the intermediates used to synthesize the final compounds were added in supplementary information file as per kind suggestion.
- The 1H and 13C NMR spectra for Intermediates I-III and Compounds 4, 5, 6, 7, 8, and 10 are missing. Please include them in the Supplementary Information.
Reply: 1H and 13C NMR spectra for Intermediates I-III and Compounds 4, 5, 6, 7, 8, and 10 were added in supplementary information file as per kind suggestion.
- Several figures display inconsistent sizing. Please standardise figure dimensions across the manuscript to ensure a uniform presentation.
Reply: Figure size was standardized as per kind suggestion.
- The Introduction would benefit from a focused rewrite. Please streamline to 4-5 concise paragraphs with a coherent progression (clinical need ‚ therapeutic limitations‚ enzyme-based rationale‚ scaffold rationale‚ study aims). Clearly articulate the research gap and novelty versus prior oxadiazole reports, and strengthen the chemical rationale (structure-property/activity arguments, with targeted citations). Conclude with a succinct objective and hypothesis that previews the assays and SAR to be tested.
Reply: Introduction was revised as per kind suggestion.
- The statement in section 3. Experimental ("washed with straight-chain hydrocarbon hexane…"), does not sufficiently describe how the products were purified or how purity was established. Kindly detail the purification workflow for each compound (e.g., silica gel flash chromatography with gradient n-hexane/EtOAc or recrystallization from EtOH, etc.)
Reply: the mentioned statement was corrected as per kind suggestion.

Reviewer 2 Report
Comments and Suggestions for Authors
The authors designed and synthesized a series of oxadiazolylbenzamide derivatives (1-10) that inhibit α-amylase and α-glucosidase. These scaffolds are intended to enhance therapeutic efficacy and reduce side effects. All derivatives were identified by spectroscopy technique and screened against α-amylase and α-glucosidase for testing their potential by comparing to the reference antidiabetic drug acarbose (4.50 ± 0.20 μM and 4.90 ± 0.30 μM, respectively). Among all the analogs and standard drugs tested, derivative 3 proved to be the most promising candidate, exhibiting the most potent inhibitory effect (IC50 = 3.20 ± 0.20 μM and 3.60 ± 0.10 μM, respectively). To further molecular docking were validated by silico studies and demonstrated strong and potent interactions between the enzymes and most of the active compounds. Density functional theory (DFT) calculations verified the electronic configuration, stability, and reactivity of the lead molecule. In addition, the ADMET spectrum predicted its good drug similarity and low toxicity. Molecular dynamics analysis further confirmed the docking results. The pharmacophore model of analog 3 supports the formation of a stable hydrogen bond network between the derivative and the enzyme receptor site. Computer simulations and in vitro experimental results jointly emphasize the therapeutic potential of these derivatives in the future effective treatment of diabetes. However, inappropriate and formatting errors in the text description may cause misunderstandings and questions to readers, and the content can also be supplemented. Therefore, I suggest accepting this paper after minor revisions. Some corrections and suggestions are listed below:
- In Figure 8 is used twice and needs to be renumbered.
- In Figure 17, some labels are missing and should be updated.
- In Table 1, the data for the standard drug acarbose appears to be offset.
- The paper only presents RMSD graphs for molecular dynamics simulations. I recommend providing additional simulation data, such as RNSF and hydrogen bond analysis.
- NMR and mass spectral data should be provided for all reported compounds.
- The manuscript contains numerous grammatical errors, formatting issues, and inconsistent capitalization. It requires further review.
- Please ensure that all compound numbers (1–10) in the text are formatted according to the journal style. For example, “(1–10)” should appear in boldface as (1–10). In addition, the compound numbers in the tables should also be presented in boldface for consistency.
- Figures should be renumbered to avoid duplication, as “Figure 8” is used twice (once for “Derivative 3 as competitive inhibitor against the diabetes enzymes” and once for “Percent inhibition of analog 3 with varied concentration for alpha-amylase complex”).
- The Introduction provides excessive detail on the classification and epidemiology of diabetes (T1D, T2D, GDM), much of which is not directly relevant to the compounds investigated in this study. It is recommended to streamline this section and focus more on the clinical need for α-amylase/α-glucosidase inhibition and related therapeutic strategies.
- The manuscript contains numerous grammatical errors, formatting issues, and inconsistent use of capitalization. Extensive language editing is required to meet the standards of Pharmaceuticals.
Extensive language need to edit by a netive english speaker.
Author Response
The authors designed and synthesized a series of oxadiazolylbenzamide derivatives (1-10) that inhibit α-amylase and α-glucosidase. These scaffolds are intended to enhance therapeutic efficacy and reduce side effects. All derivatives were identified by spectroscopy technique and screened against α-amylase and α-glucosidase for testing their potential by comparing to the reference antidiabetic drug acarbose (4.50 ± 0.20 μM and 4.90 ± 0.30 μM, respectively). Among all the analogs and standard drugs tested, derivative 3 proved to be the most promising candidate, exhibiting the most potent inhibitory effect (IC50 = 3.20 ± 0.20 μM and 3.60 ± 0.10 μM, respectively). To further molecular docking were validated by silico studies and demonstrated strong and potent interactions between the enzymes and most of the active compounds. Density functional theory (DFT) calculations verified the electronic configuration, stability, and reactivity of the lead molecule. In addition, the ADMET spectrum predicted its good drug similarity and low toxicity. Molecular dynamics analysis further confirmed the docking results. The pharmacophore model of analog 3 supports the formation of a stable hydrogen bond network between the derivative and the enzyme receptor site. Computer simulations and in vitro experimental results jointly emphasize the therapeutic potential of these derivatives in the future effective treatment of diabetes. However, inappropriate and formatting errors in the text description may cause misunderstandings and questions to readers, and the content can also be supplemented. Therefore, I suggest accepting this paper after minor revisions. Some corrections and suggestions are listed below:
- In Figure 8 is used twice and needs to be renumbered.
Reply: Figure number was corrected as per kind suggestion.
- In Figure 17, some labels are missing and should be updated.
Reply: Figure 17 illustrates the ADMET analysis profile of analogue 3. The data points labeled as “0” are not indicative of missing or incomplete data; rather, they represent parameters where the compound fell within the threshold limits defined by the predictive model. These values were retained to provide a complete and transparent overview of the computational results, ensuring that all predicted pharmacokinetic and toxicity parameters are accurately represented.
- In Table 1, the data for the standard drug acarbose appears to be offset.
Reply: The values reported for the standard drug acarbose in Table 1 are accurate and reflect the results obtained under our specific experimental conditions. While these values differ somewhat from those reported in the literature, such variations are commonly observed across studies due to differences in assay methodologies, enzyme sources (microbial or mammalian origin), substrate types and concentrations, incubation times, and temperature or pH conditions. These experimental parameters can significantly influence the apparent inhibitory potency (IC₅₀ values) of standard compounds. Therefore, the deviation does not indicate an error but rather reflects the intrinsic variability associated with in vitro enzyme inhibition assays.
- The paper only presents RMSD graphs for molecular dynamics simulations. I recommend providing additional simulation data, such as RNSF and hydrogen bond analysis.
Reply: RMSF and hydrogen bond analysis was added in the revised manuscript as per kind suggestion.
- NMR and mass spectral data should be provided for all reported compounds.
Reply: NMR and mass spectral data were provided for all reported compounds as per kind suggestion.
- The manuscript contains numerous grammatical errors, formatting issues, and inconsistent capitalization. It requires further review.
Reply: All the grammatical errors and formatting issues were removed throughout the manuscript as per kind suggestion.
- Please ensure that all compound numbers (1–10) in the text are formatted according to the journal style. For example, “(1–10)” should appear in boldface as (1–10). In addition, the compound numbers in the tables should also be presented in boldface for consistency.
Reply: Compound number was written bold throughout the manuscript as per kind suggestion.
- Figures should be renumbered to avoid duplication, as “Figure 8” is used twice (once for “Derivative 3 as competitive inhibitor against the diabetes enzymes” and once for “Percent inhibition of analog 3 with varied concentration for alpha-amylase complex”).
Reply: Figure number was revised as per kind suggestion.
- The Introduction provides excessive detail on the classification and epidemiology of diabetes (T1D, T2D, GDM), much of which is not directly relevant to the compounds investigated in this study. It is recommended to streamline this section and focus more on the clinical need for α-amylase/α-glucosidase inhibition and related therapeutic strategies.
Reply: Introduction was revised as per kind suggestion.
- The manuscript contains numerous grammatical errors, formatting issues, and inconsistent use of capitalization. Extensive language editing is required to meet the standards of Pharmaceuticals.
Reply: All the grammatical errors and formatting issues were removed throughout the manuscript as per kind suggestion.

Round 2
Reviewer 1 Report
Comments and Suggestions for Authors
I confirm that the authors have satisfactorily addressed all of my comments and made the required revisions. The manuscript is now suitable for publication.
Kind regards,